# Can multitrophic interactions shape morphometry, allometry, and fluctuating asymmetry of seed-feeding insects?

**Tamires Camila Talamonte de Oliveira**[1]*, **Angelo Barbosa Monteiro**[1†], **Lucas Del Bianco Faria**[2]

1 Graduate Program in Applied Ecology, Federal University of Lavras, Lavras, Brazil, 2 Department of Ecology and Conservation, Institute of Natural Science, Federal University of Lavras, Lavras, Brazil

† Deceased.

* tamires_talamonte@hotmail.com

**Data Availability Statement:** All relevant data are within the manuscript and its Supporting Information files.

## Abstract

Body size is commonly associated with biological features such as reproductive capacity, competition, and resource acquisition. Many studies have tried to understand how these isolated factors can affect the body pattern of individuals. However, little is known about how interactions among species in multitrophic communities determine the body shape of individuals exploiting the same resource. Here, we evaluate the effect of fruit infestation, parasitism rate, and seed biomass on size, allometric and asymmetric patterns of morphological structures of insects that exploit the same resource. To test it, we measured 750 individuals associated with the plant *Senegalia tenuifolia* (Fabaceae), previously collected over three consecutive years. Negative allometry was maintained for all species, suggesting that with increasing body size the body structure did not grow proportionally. Despite this, some variations in allometric slopes suggest that interactions in a multitrophic food web can shape the development of these species. Also, we observed a higher confidence interval at higher categories of infestation and parasitism rate, suggesting a great variability in the allometric scaling. We did not observe fluctuating asymmetry for any category or species, but we found some changes in morphological structures, depending on the variables tested. These findings show that both allometry and morphological trait measurements are the most indicated in studies focused on interactions and morphometry. Finally, we show that, except for the fluctuating asymmetry, each species and morphological structure respond differently to interactions, even if the individuals play the same functional role within the food web.

## Introduction

During their development, organisms are exposed to numerous stresses, such as severe weather conditions [1], pollution [2], deficiency of nutrients [3], competition [4], predation and parasitism [5, 6]. These stresses can directly affect the body size of organisms [7], which in turn can influence changes in their fertility, survival, and dispersion [8–10]. When organisms

**Funding:** TCTO thanks the Brazilian Coordination for the Improvement of Higher Education Personnel (CAPES) and the National Council for Scientific and Technological Development (CNPq) (141129/2018-2) for financial support. LDBF thanks CAPES, CNPq (306196/2018-2), and Fundação de Amparo à Pesquisa do Estado de Minas Gerais (FAPEMIG) for financial support. The funders had no role in study design, data collection and analysis, decision to publish, or preparation of the manuscript.

**Competing interests:** The authors have declared that no competing interests exist.

suffer stresses above their tolerance, they need to compensate for the loss of energy, reducing the amount of energy allocated for the growth, maintenance, and development of the morphological structures, which produces distortions in development, body size and symmetry [11–16]. However, some organisms can withstand stress through "buffering" mechanisms, producing pre-determined phenotypic characteristics, which limit the variation of the phenotype and reduce the effects of stress on their development [17].

Two tools commonly used to evaluate possible phenotypic responses caused by stress are the allometry and fluctuating asymmetry (FA) [15, 18]. Static allometry is used to evaluate the mechanisms that influence variations in the growth of co-specific individuals in the same stage of life. The relationship between individuals' development is observed considering body size and its organs, or between two organs (e.g. scaling relationship) [19–23]. FA refers to small random deviations from the symmetry of bilaterally symmetrical traits and it can be used as an indicator of the individual's ability to maintain a symmetrical development in the face of biotic and abiotic disorders [16, 24].

In recent years, with technological advances, there has been a significant increase in morphometric investigations, especially in the group of insects. This has directly reflected in a better understanding of the influence of abiotic and biotic factors on the morphological aspects of these individuals [16, 19, 23, 25–28]. Nevertheless, most studies have analyzed isolated factors influencing the body size of organisms [23, 27–33] and few considered multiple interactions affecting body size at the same time [34, 35]. Recently, we observed that interactions in a multi-trophic network can drive to changes in the body size of seed-feeding insects differently, in which abundant species have their body size more affected than less abundant ones [35]. Likewise, although there are several studies considering isolated factors influencing morphometric patterns (e.g. allometry and FA) [16, 20, 23, 25, 36], we are not aware of previous studies that have assessed how multiple interactions may or may not affect allometric patterns and FA of insects that share the same resource, especially in natural environments.

Thus, the present study evaluates the effect of different interactions on the morphology of three different insects, which exploit the same plant resource. To do that, we assessed the allometry and fluctuating asymmetry of the beetles *Merobruchus terani* Kingsolver, 1980, and *Stator maculatopygus* (Pic, 1930) (Chrysomelidae: Bruchinae) and the wasp *Allorhogas vulgaris* Zaldívar-Riverón & Martinez [37] (Hymenoptera: Braconidae: Doryctinae). These three species of seed-feeding insects are associated with the fruits of the plant *Senegalia tenuifolia* (L.) Britton & Rose (Fabaceae: Mimosoideae). Their immature stages live and feed inside the seed and when they reach adulthood, they can feed on nectar and pollen [38–40]. Moreover, because they spend much of their lives inside the seed, and their mobility is limited by their life history, these insects can experience high levels of competition and parasitism. Furthermore, their morphological characteristics are closely related to the features of their resource [27, 28, 33, 41, 42]. These biological aspects make this system an advantageous study model for understanding how interactions can affect insect morphometric patterns in a natural system.

We hypothesize that an increase in fruit infestation, parasitism rate, and a reduction in seed biomass cause: (I) a negative change in allometric slopes; (II) greater deviations in the fluctuating asymmetry of these species; (III) and reduction in the size of their morphological structures. To test these hypotheses, we selected the morphological traits that best explain the body size variation in some species of bruchine beetles [20, 23], namely pronotum length, left and right elytra and biomass; and for the wasp: right and left wing, right and left tibia length, and total body length of *A. vulgaris* [43, 44]. We evaluated the size, the allometric relationships and the fluctuating asymmetry of these morphological traits, according to different categories of seed infestation, parasitism rate and seed biomass to investigate the possible influence of these variables on the morphological structures of these species.

## Materials and methods

### Field location

We carried out this study at the municipalities of Lavras and Luminárias, in fragments of Brazilian Cerrado (savanna) in the south of Minas Gerais state, south-eastern Brazil. The study areas were divided into three main fragments called Ae, La, and Lu, which were 6 km apart from each other. Across these three areas, we established seven subareas which were at least 400 m from each other (Ae-1: 21°14′5.71″S- 44°57′8.66″W; Ae-2: 21°14′7.87″S-44 58′0.06″W; La-1: 21°-18′3.46″S- 44°58′0.53″W; Lu-1: 21 31′1.36″S-44°53′1.78″W; Lu-2: 21 31′5.13″S-44°52′6.32″W; Lu-3: 21 31′5.31″S-44 52′3.84″W; and Lu-4: 21°41′9.88″S- 44°96′7.18″W); see [45] for additional details of the study site.

### The plant system, the assessment of fruits, seed biomass and insects

No permits or authorization were obtained as Brazilian authorities did not require permits for collection by public roads. *Senegalia tenuifolia* is locally known as *unha-de-gato*, and it is a pioneer species largely distributed in South America [46]. *Senegalia tenuifolia* is characterized as a scandent shrub, i.e., a shrub that is able to become a liana and reach a greater height (up to 8 m) [47]. The flowering period occurs from November to January, and the ripening occurs from January to August (Faria, L.D.B., personal communication). After this phase, fruits open and seeds begin to fall on the ground [45].

The *S. tenuifolia* fruits were previously collected in 2012, 2013, and 2014 in June, July, and August, during the plant's ripening season. In this stage, plants can be recognized by their brownish color. Nevertheless, we only considered the collections of the months of July and August, as insects collected in June were not developed, making insect identification impossible. As *S. tenuifolia* is a liana, we could not distinguish the number of different plants at each site, so we opted to collect the fruits randomly in each subarea [33]. We collected 25 fruits from seven subareas for each month; thus, for each year, we had a sample size of 350 fruits except for 2013 (349 fruits sampled) totaling 1,049 fruits. Furthermore, there were different numbers of sampling sites for each area (see Table 1 for more details). All fruits were collected when still attached to the mother plant and taken to the laboratory, where each fruit was stored in individual PVC tubes covered with voile to enable air circulation. The fruits were stored for three months to allow the insects inside the seeds to complete their development and emerge as adults. After this, we opened the fruits and assessed the seed biomass by separating them into paper bags and drying at 40°C for 48 h. Subsequently, they were weighed in precision analytical balance in order to obtain the dry biomass.

The emerged insects of each fruit were identified at the genus level and species level when possible, stored in 1.5-mL labeled plastic microtubes containing 70% ethanol and deposited in the Entomological Collection of the Laboratory of Ecology and Complexity at the Federal

**Table 1. Sampling site areas with the amount of fruits gathered per subareas in each year.**

| Sampling sites and number of fruits per area in each year | | | |
|---|---|---|---|
| **Areas** | **2012** | **2013** | **2014** |
| **Ae** | Ae1, Ae2 | Ae1, Ae2 | Ae1, Ae2 |
| | 100 fruits | 99 fruits | 100 fruits |
| **La** | La1 | La1 | La1 |
| | 50 fruits | 50 fruits | 50 fruits |
| **Lu** | Lu1, Lu2, Lu3, Lu4 | Lu1, Lu2, Lu3, Lu4 | Lu1, Lu2, Lu3, Lu4 |
| | 200 fruits | 200 fruits | 200 fruits |

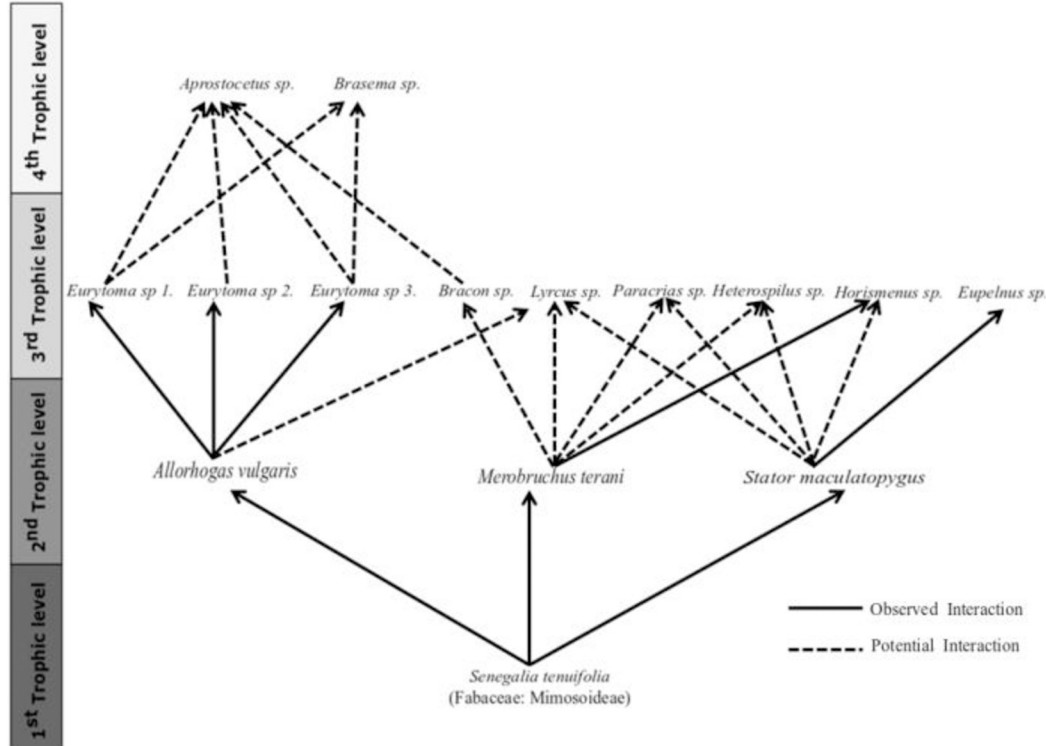

**Fig 1. The *Senegalia tenuifolia* and related trophic levels.** The resource plant represents the first trophic level, the second trophic level comprises the three main seed-feeding species, the third trophic level comprises the parasitoid species, and the fourth trophic level comprises the hyperparasitoid species. Solid lines indicate observed interaction and dashed lines indicate potential interaction. Potential interactions were based on information from previous studies. Figure obtained from previous studies [35].

University of Lavras, Minas Gerais, Brazil. Subsequently, we used these specimens to measure their morphological structures. Building upon data previously collected, we gathered information on seed biomass of *S. tenuifolia* and on the abundance of the three seed-feeding species (*M. terani* Kingsolver, 1980, *S. maculatopygus* (Pic, 1930) and *A. vulgaris* Zaldívar-Riverón & Martínez 2018), and their parasitoids in each fruit [45, 48]. We chose these three species because they are the most abundant species consuming *S. tenuifolia*. The simplified food web is displayed in Fig 1.

## Morphological measurements

Specimens of *M. terani*, *S. maculatopygus*, and *A. vulgaris* were placed on slides in the dorso-ventral position for microscopy. The slides were photographed using a Leica M205A stereo-scopic microscope coupled to a Leica DFC295 camera. Subsequently, measurements of the morphological structures were made based on the photos uploaded in Leica application suite. The structure measurements in the two coleopteran species were pronotum width, and both left and right elytra length and width [20, 23] (Fig 2A and 2B). For the hymenopteran species *A. vulgaris*, we measured total body size length (thorax + abdomen) [43, 49, 50], +R vein until the end of 3RSb vein, wing width measured from the junction between anterior wing length measured from the beginning of C+Sc, the parastigma and stigma to the end of vein 1A (Fig 2C) (adapted from [44]); and posterior tibia length was measured from the junction of the femur and the tibia to the junction of the tibia and tarsus [23, 35]. Wings and tibia from the

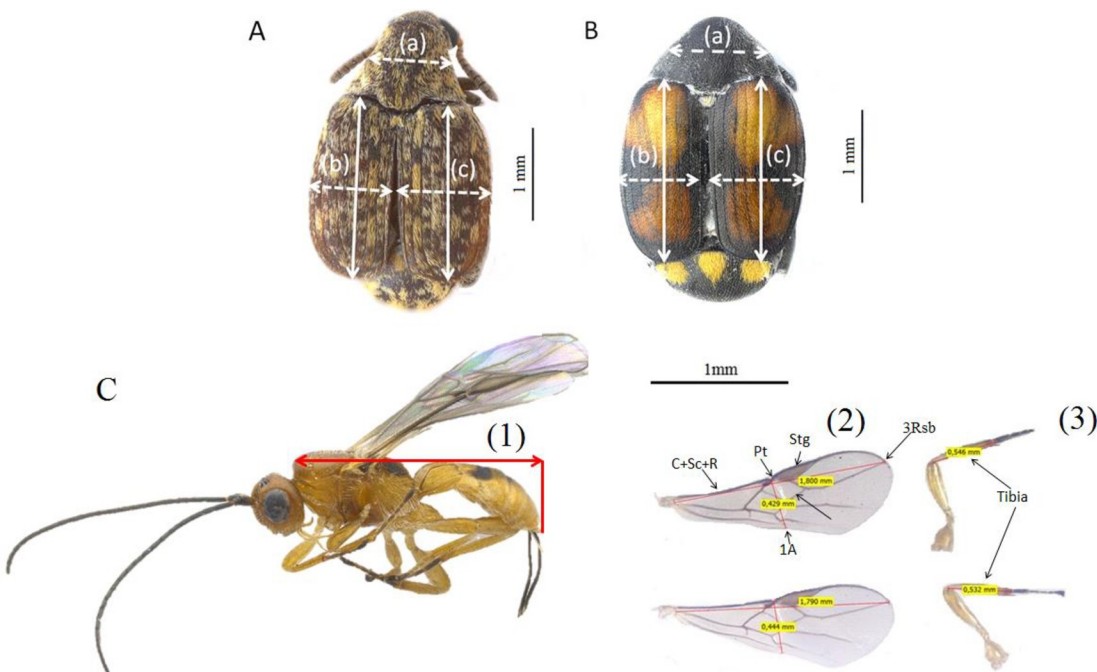

**Fig 2.** Body traits of the seed-feeding beetles (A) *Merobruchus terani* and (B) *Stator maculatopygus* (Chrysomelidae: Bruchinae) and the wasp (C) *Allorhogas vulgaris* (Hymenoptera: Braconidae: Doryctinae) taken to estimate morphometry, allometry, and fluctuating asymmetry of their morphological structures, according to resource size, fruit infestation, and parasitism rate. The solid arrows indicate the length and the dashed arrows the width of (a) pronotum, (b) left elytra, (c) right elytra for the beetles, and (1) indicates the measurement of total body size length (thorax + abdomen); (2) the forewing length measured from the beginning of C+Sc+R vein until the end of 3RSb vein, forewing width measured from the junction between the parastigma (Pt) and stigma (Stg) to the end of vein 1A; and (3) posterior tibia length for the wasp.

wasp *A. vulgaris* individuals were dissected from their bodies to reduce measurement error; we measured both the right and left sides (Fig 2C). All measurements were performed three times.

## Data analysis and categories

We tested the effect of fruit infestation, parasitism and resource size (seed biomass) on morphometric patterns of the three seed-feeding species. To do it, we calculated the fruit infestation rate (FIR) (1) and fruit parasitism rate (FPR) (2) by the following formulae:

$$\text{FIR} = \frac{HT}{SA} \tag{1}$$

$$\text{FPR} = \frac{PT}{HT + PT} \tag{2}$$

Where HT is the total abundance of herbivorous, SA is the total number of seed per fruit, PT is the total number of parasitoids in the fruit, and HT is the total abundance of seed-feeding insects in the fruit (i.e., *Merobruchus terani*, *Stator maculatopygus* and *Allorhogas vulgaris*).

We defined three categories for both fruit infestation and parasitism rate: "Low", from 0 to 0.30; "Medium", from 0.31 to 0.60; and "High" from 0.61 to 1 (adapted from [43]). In addition, to have an approximately equal amount of samples by each category of seed biomass, we divided them into: "Small"; from 0.025 mg to 0.266 mg; "Medium" from 0.271 mg to 0.454 mg; and "Large" from 0.454 mg to 1. 993 mg.

Preliminary Spearman correlation analysis indicated that the elytra length and width of *M. terani* (correlation of 0.88) and *S. maculatopyus* (correlation of 0.77), as well as the anterior wing length and width of *A. vulgaris* (correlation of 0.89) were highly correlated. Thus, the morphometric analyses were conducted using only the length of these morphological structures.

Allometry was evaluated according to categories of fruit infestation, parasitism rate and seed biomass, estimating the rate of variation of the pronotum and elytra (mean of the right and left sides) of *M. terani* and *S. maculatopygus* in relation to the individual body weight, for *A. vulgaris*, we estimated the rate of variation of the wing and tibia (mean of the right and left sides) in relation to the total body size of each individual. In this analysis, we used 'major axis' regressions (i.e., geometric mean), which are considered more appropriate when both x and y variables are measured with error [23, 51]. The values of slope and confidence intervals from the analyses were used in a single plot to illustrate the allometric relationships among species and all categories.

We evaluated the FA patterns of species according to categories of fruit infestation, parasitism rate, and seed biomass. We selected the following morphological structures: elytra length (right and left) of *M. terani* and *S. maculatopygus* and wing and tibia length (right and left) of *A. vulgaris*. We used generalized linear mixed models (GLMM) analysis with restricted likelihood (REML), which produces unbiased estimates for the values of fluctuating asymmetry (i.e., fixed effects), considering possible measurement errors among individuals (i.e., random effects) [52]. The fixed effects evaluate directional asymmetry (DA), while the random effects evaluate fluctuating asymmetry (FA). All models considered random terms for the intercept, which estimates an average value among the three individual measurements. Besides, the structure of the random effect for the slope evaluates FA, estimating the same growth rate between the sides (M1), or different growth rates between the sides (M2). We also considered variations in the slope between the sides associated with the categories of infestation rate (M3) and parasitism rate (M4). We selected the best model through the likelihood ratio test (see supporting information S1 Table). Considering the best model for the structure of random effects (RE), we also evaluated the variation in the size of the structures among the categories of fruit infestation, parasitism rate, and seed biomass. Significance levels between groups were tested by post-hoc Tukey, preventing overestimated inferences with the 'lsmeans' package [53]. All analyses were performed using R software v.3.4.1 [54]. Allometry analyses were performed using the 'lmodel2' package [55]. REML analyses were conducted with the 'lme4' package [56] and 'lmerTest' package [57]. All relevant data are within the paper and its Supporting Information files.

## Results

We measured 750 seed-feeding insects, 534 of these were *M. terani* individuals, 146 were *A. vulgaris* individuals, and 70 were *S. maculatopygus* individuals, distributed according to the categories displayed in Table 2. The number of insects present in the highest categories of fruit infestation and parasitism was lower concerning the medium and low categories for all species, except for *A. vulgaris*, which was more abundant in fruits with higher infestation rate. We did not find individuals of *M. terani* or *S. maculatopygus* in fruits with a high infestation rate, and their abundances were lower in high parasitism rate (Table 2).

Regarding the allometric analysis, we found slope values less than one, which suggests negative allometric patterns for all species (Figs 3, 4 and 5). This means that the structures measured (pronotum and elytra length in the beetles, and wing and tibia length in the wasp) increased proportionally less in their length concerning their body size. Despite that, we

**Table 2. Total abundance of the three seed-feeding insects,** *Merobruchus terani*, *Stator maculatopygus* and *Allorhogas vulgaris*, **associated to the plant** *Senegalia tenuifolia* by category of fruit infestation, parasitism and seed biomass.

| CATEGORIES | *Merobruchus terani* | *Stator maculatopygus* | *Allorhogas vulgaris* |
|---|---|---|---|
| Low fruit infestation (LFI) | 323 | 45 | 34 |
| Medium fruit infestation (MFI) | 191 | 4 | 51 |
| High fruit infestation (HFI) | 0 | 0 | 61 |
| Low parasitism rate (LPR) | 432 | 43 | 114 |
| Medium parasitism rate (MPR) | 71 | 4 | 28 |
| High parasitism rate (HPR) | 15 | 3 | 0 |
| Small seed (SS) | 183 | 8 | 61 |
| Medium seed (MS) | 181 | 11 | 62 |
| Large seed (LS) | 154 | 31 | 23 |
| **Total abundance** | 534 | 70 | 146 |

observed small variations of allometric slopes and confidence intervals among the categories and species, indicating some kind of effect on their morphological pattern (See supplementary material, S2–S4 Tables).

Fruit infestation positively influenced the pronotum allometry (increased allometric scale) of *M. terani* and both wing and tibia allometry of *A. vulgaris*. The slope variation between the pronotum and body weight (pronotum allometry) in *M. terani* individuals was 0.08 in LFI (low fruit infestation), 0.16 in MFI (medium fruit infestation) (Fig 3). To *A. vulgaris*, the slope was 0.45 in LFI, 0.55 in MFI and 0.71 in HFI (high fruit infestation) between its wing and total body size (wing allometry), and 0.45 in LFI, 0.55 in MFI and 0.76 in HFI between its tibia and

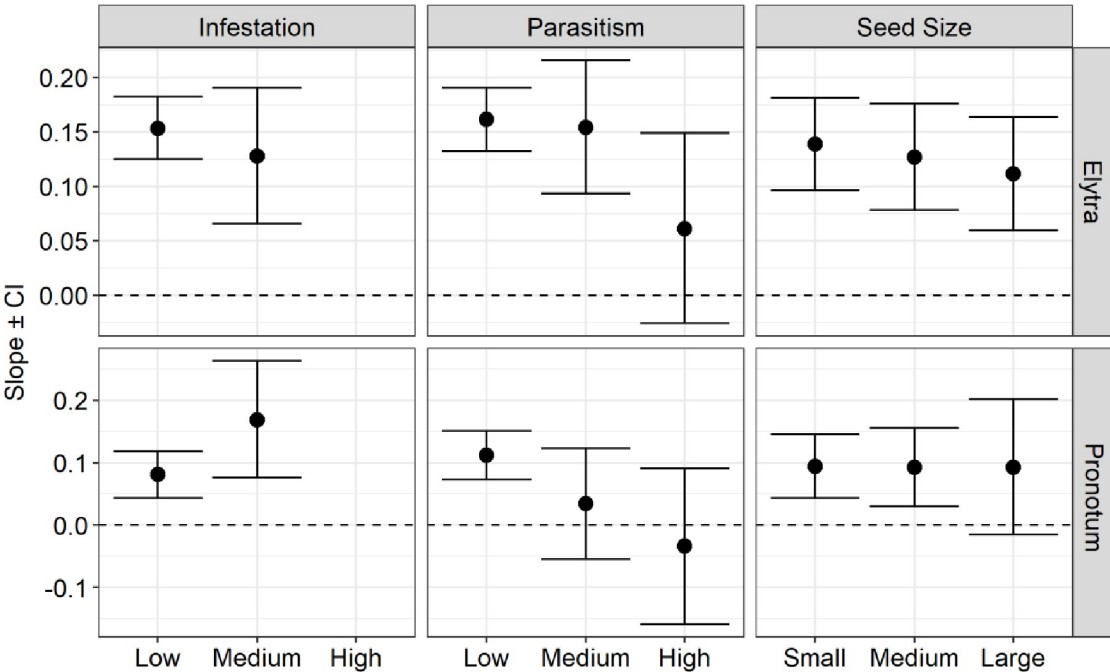

**Fig 3. Negative allometry depicted by the slopes and their confidence intervals (CI) for the pronotum and elytron allometry (pronotum length and elytron length in relation to body weight) among infestation, parasitism rate, and seedbiomass categories for** *Merobruchus terani* **individuals (CI of 95%).** Low (0–0.30%), medium (0.31–0.60%), and high (0.61–1%) infestation and parasitism rate; and among small, medium, and large seeds.

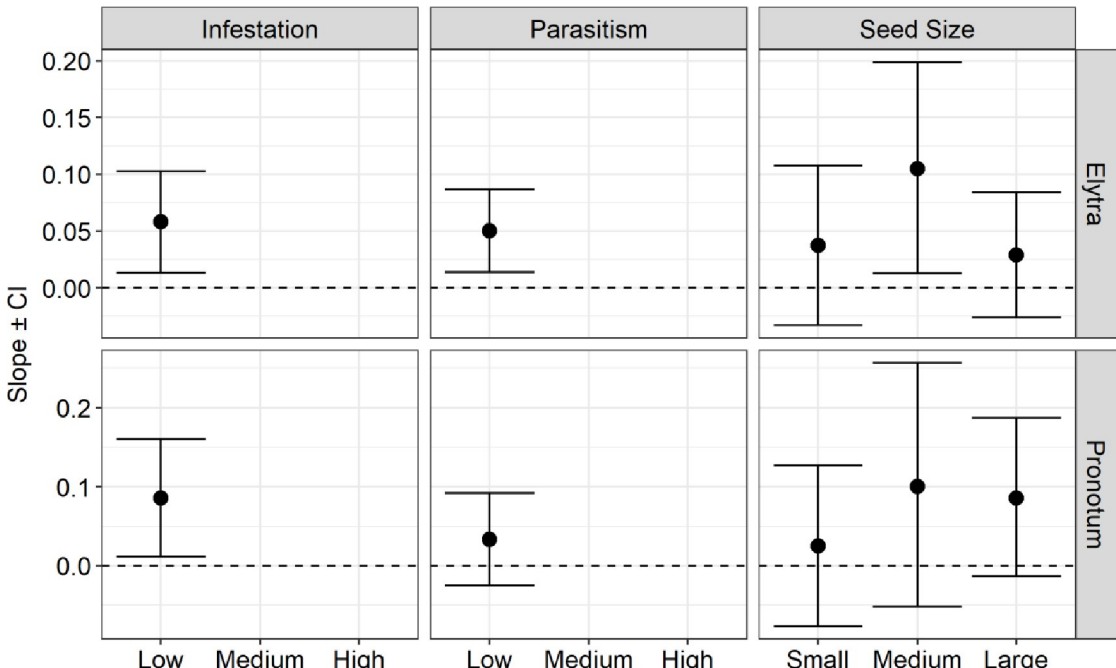

**Fig 4. Negative allometry depicted by the slopes and their confidence intervals (CI) for the pronotum and elytron allometry (pronotum length and elytron length in relation to body weight) among infestation, parasitism rate, and seed biomass categories for *Stator maculatopygus* individuals (CI of 95%).** Low infestation categories (0%-0.30%) and low parasitism categories (0%-30%), and between small, medium, and large seeds.

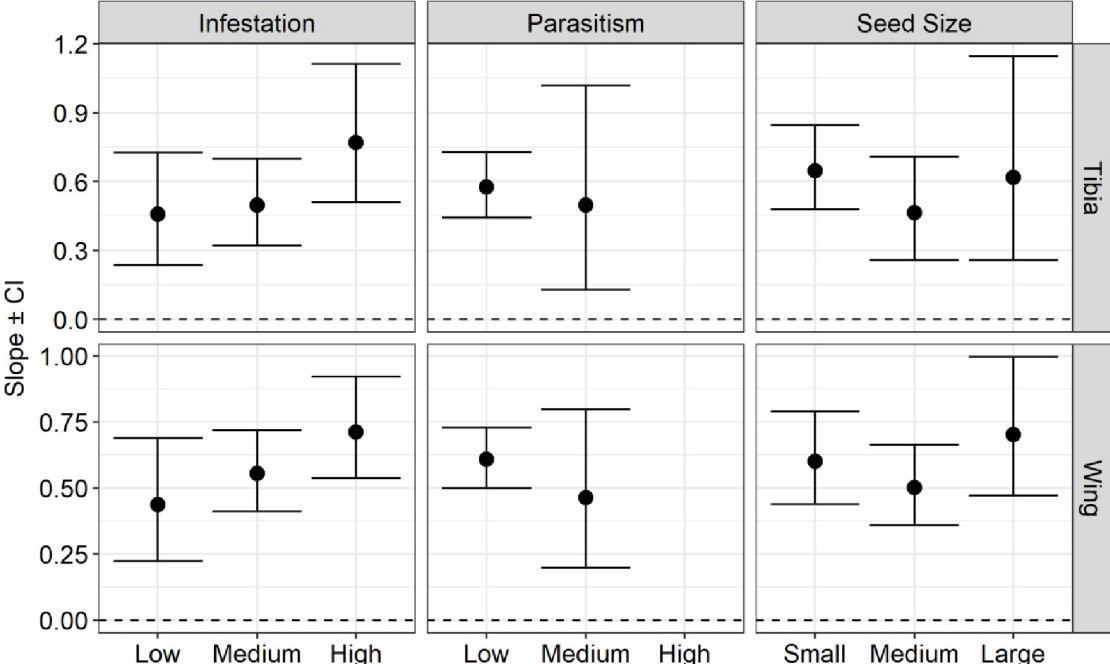

**Fig 5. Negative allometry depicted by the slopes and their confidence intervals (CI) for the tibia and wing allometry (tibia length and wing length in relation to body size) among infestation, parasitism rate, and seed biomass categories for *Allorhogas vulgaris* individuals (CI of 95%).** Low (0–0.30%), medium (0.31–0.60%), and high (0.61–1%) infestation and parasitism rate (low and medium); and among small, medium, and large seeds.

total body size (tibia allometry) (Fig 5). On the other hand, we found a weak negative effect (decreased in allometric scale) of the fruit infestation on the elytra allometry of *M terani*, with a variation of allometric slope between elytra and body weight of 0.15 in LFI; and 0.12 in MFI (Fig 3).

The rate of parasitism negatively influenced the pronotum and elytra allometry of *M. terani* individuals and the wing and tibia allometry of *A. vulgaris* individuals. The slope variations between pronotum and body weight of *M. terani* (pronotum allometry) were: 0.07 in LP (low parasitism), 0.03 in MP (medium parasitism); and 0.03 in HP (high parasitism); and between elytra and body weight of *M. terani* (elytra allometry) were 0.16 in LP; 0.25 in MP; and 0.06 in HP (Fig 3). The slope variations of the wing and total body size of *A. vulgaris* (wing allometry) were 0.60 in LP and 0.46 in MP, and between the tibia and total body size of *A. vulgaris* (tibia allometry) were LP: 0.57; MP: 0.49 (Fig 5).

Regarding the seed biomass categories, we found a slight negative variation in the elytra allometry (variation between elytra and body weight) of *M. Terani* with slope values of 0.13 in SS (small seed), 0.12 in MS (medium seed) and 0.11 in LS (large seed). No variations were found for *M terani* pronotum allometry, since the slope was 0.09 in all categories of seed biomass (SS, MS, and LS) (Fig 3). On the other hand, we found a positive variation in wing allometry of *A. vulgaris* individuals, with slopes of 0.60 in SS, 0.50 in MS, and 0.70 in LS, as well as their tibia allometry with slopes of 0.64 in SS, 0.46 in MS, and 0.61 in LS (Fig 5). For *S. maculatopygus* individuals, we found higher allometric values in MS category, both to their elytra (slope values of 0.03 in SS, 0.10 in MS, and 0.03 in LS) and pronotum (slope values of 0.03 in SS, 0.10 in MS, and 0.08 in LS) (Fig 4). Furthermore, a closer inspection on confidence intervals indicated isometry of *A. vulgaris* at high rates of infestation, parasitism, and large seeds, once confidence intervals were ≥ 1 in these situations (Fig 5).

We did not observe fluctuating asymmetry for any morphological structure tested, as the results obtained by the selecting models showed no significant differences among the models tested for all species (see supplementary material S5 Table). Thus, in all cases, we considered the simplest model (M1), which does not distinguish random effects for the slope between the right and left sides (i.e., absence of FA). Additionally, we verified a significant increase in the elytra length of *M. terani* with the increase of seed biomass and fruit infestation (Fig 6). The effect of seed biomass was significantly dependent on the category, but not on the side, which suggests that there is neither fluctuating (FA) nor directional asymmetry (DA).We also observed a significant effect of wing size of *A. vulgaris* in relation to seed biomass, regardless of the side: smaller seeds caused a greater increase in the wing size of *A. vulgaris* (see supplementary material, S7 Table) (Fig 7). We did not observe any relation of seed biomass, parasitism rate and infestation rate in the elytra length of *S. maculatopygus* (see supplementary material, S8 Table), elytra length of *M. terani* (see supplementary material S6 Table), or in the tibia length of *A. vulgaris* (see supplementary material S7 Table).

## Discussion

Through the measurement of various morphological traits of the three main seed-feeding insects associated with fruits of *S. tenuifolia*, we have demonstrated the potential influence of multiple trophic interactions on the body patterns (especially in allometry and morphometry) of seed-feeding insects that share the same resource. We tested this by using different categories of fruit infestation, parasitism rate, and resource size. In general, we found a negative allometric pattern for all species with some variations in allometric slopes, suggesting that interactions in a multitrophic food web can indeed shape the development of these insects' bodies. Moreover, we observed a substantial increase in confidence intervals at high levels of

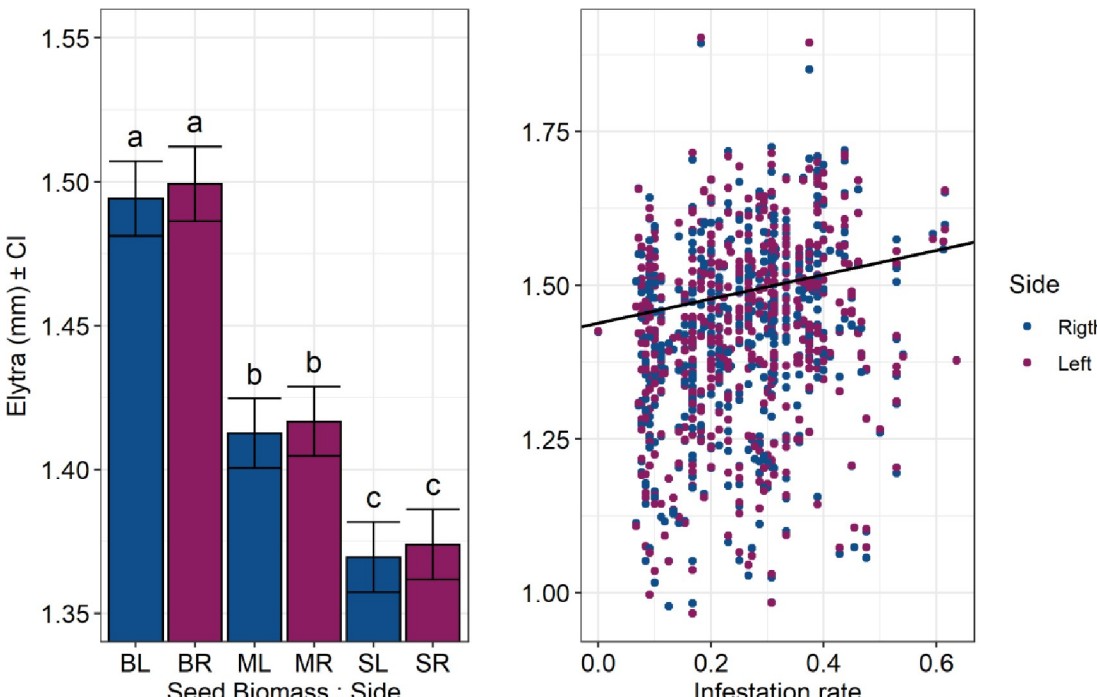

**Fig 6. Fluctuating asymmetry and the effect of seed biomass and infestation rate on the elytra length of *Merubruchus. terani*.**
The effect was estimated using mixed linear models adjusted by the REML method; the result compares the right and left sides of the elytra. The Y- axis represents the elytra length of the *M. terani* and the X-axis represents the seed biomass according to the side evaluated and infestation rate: BL (large seed, left elytron), BR (large seed, right elytron), ML (medium seed, left elytron), MR (medium seed, right elytron), SL (small seed, left elytron), and SR (small seed, right elytron). Different letters indicate statistically different means between categories.

infestation and parasitism, which suggests that, at high intensity, stresses can lead to an increase in variations of the three seed-feeding insects' bodies, changing their allometric scaling, but not their allometric pattern (slope values < than 1). We did not observe fluctuating asymmetry for any of the species studied; however, we observed a significant increase in the elytra of *M. terani* at high infestation and a reduction in wing length of *A. vulgaris* in large seeds.

In the group of insects, both body size and locomotion structures (e.g. wing and tibia) are essential for dispersion, competitive ability, and escape from predators [10]. The scale between individuals' body sizes and their morphological structures can be changed due to strong environmental and genetic pressures during their development [58, 59]. However, to avoid abrupt changes affecting the performance of individuals, there is a certain flexibility in the growth of body size and its morphological structures [60]. We observed this flexibility in the allometric pattern of all the seed-feeding insect species studied that despite maintained a negative pattern regardless the categories tested, produced greater variations in morphological traits at high levels of fruit infestation and parasitism.

The fruit infestation, parasitism rate, and biomass of the resource did not change the morphological allometric pattern of the three seed-feeding insect species. However, as we found a great variation in allometric slopes, it seems these variables (e.g. fruit infestation, fruit parasitism and seed biomass) can influence the insect's development, both in its allometry and abundance. Therefore, since these variables negatively changed (decreased) some allometric scaling of these three seed-feeding species, our first hypothesis was partially corroborated. The fruit infestation and seed biomass caused a positive (increased allometric slopes) variation in the

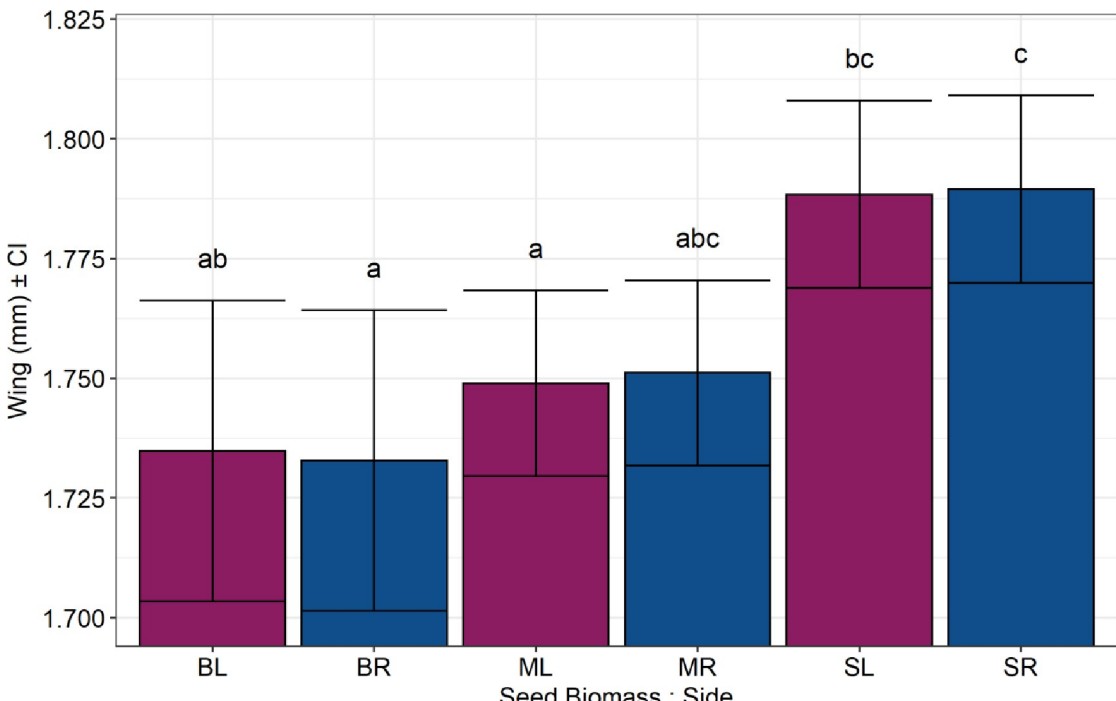

**Fig 7. Fluctuating asymmetry and the effect of seed biomass on the wing length of *Allorhogas vulgaris*.** The effect was estimated using mixed linear models adjusted by the REML method; the result compares the right and left sides of the wing. The Y-axis represents the wing length of *A. vulgaris* and the X-axis represents the seed biomass: BL (large seed, left wing), BR (large seed, right wing), ML (medium seed, left wing), MR (medium seed, right wing), SL (small seed, left wing), SR (small seed, right wing). The purple color represents the left side and the blue color represents the right side. Different letters indicate statistically different means among categories.

allometry of the pronotum of *M. terani* and the allometry of wings and tibia of *A. vulgaris*. Studies have shown that the amount of resource available during the larval phase can affect the relationship between body size and morphological structures of some individuals [5, 58, 60–63]. In one study, for instance, they observed that insects that did not have enough food during their larval stage had their body size reduced and wings proportionally larger [60]. This happened because the initial lack of food prevented the body size of individuals from growing, whereas the wings continued to grow exponentially, increasing their wing allometric coefficient. Also, a continuous growth in appendages of holometabolous insects was observed, even after their body size had stopped growing [64]. Thus, the increase of fruit infestation, and consequently, the decrease in resource availability would be affecting the body weight of these individuals, without necessarily leading to a decrease in their morphological structures, increasing the allometric slope of their morphological structures.

On the other hand, the infestation categories and seed biomass negatively influenced (decreased) the elytra allometry slopes of *M. terani*. This result is similar to the one obtained in another study, in which it was observed that different morphological structures in the same individual can present different plasticity in face of biotic and abiotic factors [59]. Therefore, as the same factor (e. g. fruit infestation and seed biomass) can have a different effect on different morphological traits, we strongly recommend measuring more than one morphometric structure in the same individual, especially in morphometric studies.

The parasitism rate negatively changed the allometric pattern of both the pronotum and elytra of *M. terani*, and the wing and tibia of *A. vulgaris*. This may have occurred because the

body size and morphological structure growth in holometabolous insects occur at different times. First, there is an increase in body size, which is proportional to the amount of food available during the larval phase. After that, individuals stop eating and enter the pupal stage, when their morphological structures start the development process [65]. Insects exposed to the presence of parasitoids can change their foraging habitat inside fruits [66, 67]. In one study, it was found that both insect hosts and their parasitoids can detect the presence of each other by vibrations emitted when they are foraging [66]. The parasitoids can detect their host by the vibration emitted while they are feeding on the seeds, and the host can detect the presence of parasitoids by the vibration they emit when they insert their ovipositor inside the fruit. Consequently, to survive, these hosts stop eating or accelerate their development [68, 69]. Thus, although the morphological structures grow independently from body size, both the lack of food and the acceleration of their development could have caused these changes in the allometric scale of these seed-feeding insect species.

Furthermore, we were unable to observe the effect of the infestation and parasitism rate on the allometry of the structures of *S. maculatopygus*, as they were absent in the medium and high categories, showing somehow their low tolerance to stresses caused by fruit infestation and parasitism, something already suggested in other studies [23, 33, 35]. Besides, the values of the elytra and pronotum allometry of *S. maculatopygus* were close to zero, indicating that the allometry of *S. maculatopygus* is not a very sensitive parameter to changes in the amount of the resource, which can be reinforced by the results of other studies, in which the body size of *S. maculatopygus* was not related to the size of the resource [23, 35].

In addition to allometry pattern, another approach used in morphometric studies is FA. Some studies consider FA to be a good indicator of instability in the development of individuals under stressful situations [1, 12, 70–72]. However, other studies failed to find FA responses under stressful situations caused by larval density [73] and lack of food [74, 75]. Likewise, we did not find differences between the right and left sides (e.g., FA) of the three seed-feeding insects, regardless of the categories tested (e.g., infestation rate, parasitism rate, and seed biomass), rejecting our hypothesis that greater stresses would cause greater deviations in the symmetry of the three seed-feeding insect' species. Some studies have suggested that body size is more sensitive to environmental disturbances than FA, since body size of individuals can present a greater plasticity than FA [75–77]. Therefore, due to the great plasticity of body size, there is a dampening of the impact caused by stressful factors in the development of organisms, thus maintaining their shape, symmetry, and, consequently, performance.

Our results confirm that both body size and morphological structure size are more plastic (higher variability) than changes in their symmetry (FA), since we verified some significant relationships between the measured body structures and the categories (e.g., competition and size of the resource), regardless of the side evaluated. The seed biomass showed a positive relationship with the elytra size of *M. terani*, and a negative one with the wing size of *A. vulgaris*. Besides, we found that, at high infestation rate, *M. terani* elytra's size is bigger. The increase in the *M. terani* elytra was also observed in another study, in which it was verified that increase of competition generates both bigger elytra and body size for this species [23]. By contrast, the results obtained by *A. vulgaris* indicate that larger fruits cause a decrease in their wings. However, this result may reflect the habit of this species to feed on the ends of the seed, allowing the exploitation of the resource by more than one individual [78]. In addition to this, we observed in the laboratory that the seeds preyed upon by these individuals are almost completely consumed, which would explain this negative relationship between the seed biomass and the body size of *A. vulgaris* [35].

We did not find significant influences of fruit infestation and parasitism on the morphological traits of any of the three species, refuting our hypothesis that higher levels of infestation

and parasitism would cause a decrease in the morphological structures of these species. Also, the hypothesis that a decrease in the amount of resource would decrease the growth of morphological structures was only corroborated for *M. terani*. These results show that although fruit infestation and parasitism are limiting factors on the abundance of these individuals at higher rates, it is not necessarily reflected in changes in their symmetry, but instead can cause changes in the size of morphological structures, indicating that this approach would be most interesting in studies focusing on interaction and morphometry.

In conclusion, we have demonstrated for the first time the influences of fruit infestation, parasitism rate, and resource size on the body shape of three seed-feeding insect species that share the same resource. We did that by analyzing their allometry, symmetry, and morphometry. We found that although these species have a similar functional role (consuming seeds) in the food web, their bodies have different ways to respond to these interactions. Also, although their body scaling did not change its pattern, keeping slope values <1, the relationship between body size and morphological structures varies substantially at high levels of infestation and fruit infestation (higher CI values), demonstrating that this tool can be effectively used in evaluations of this type. Last, we suggest more studies evaluating the relationship between FA and stresses caused by interactions within a food web, since it was not clear if there was no effect on FA, or if body parts changed first.

## Supporting information

**S1 Table. Model testing the existence or not of fluctuating asymmetry with different combinations of random effects for the intercept and slope, by testing mixed models with restricted likelihood (REML).**
(DOCX)

**S2 Table. Allometric coefficient with slope value, confidence interval according to the categories and morphological structures of *Merobruchus terani*.**
(DOCX)

**S3 Table. Allometric coefficient with slope value, confidence interval according to the categories and morphological structures of *Stator maculatopygus*.**
(DOCX)

**S4 Table. Allometric coefficient with slope value, confidence interval according to the categories and morphological structures of *Allorhogasvulgaris*.**
(DOCX)

**S5 Table. Result comparing the fluctuating asymmetry models of *Merobruchus terani* and *Stator. maculatopygus* elytra and *Allorhogas vulgaris'* wing and tibia.** All models have the same structure with different random factors. In which, M1 is the model with no difference in sides; M2 with different slopes for each side; M3 with different slopes for each side in infestation categories; M4 different slopes for each side in parasitism categories.
(DOCX)

**S6 Table. Fluctuating asymmetry between left and right sides for elytra length of *Merobruchus terani*, according to categories of seed biomass, fruit infestation and parasitism rate.** Results are displayed in comparison to the right elytra.
(DOCX)

**S7 Table. Fluctuating asymmetry between left and right sides for wing and tibia length of *Allorhogas. vulgaris*, according to categories of seed biomass, fruit infestation and**

**parasitism rate. Results are displayed in comparison to the right side of these structures.**
(DOCX)

**S8 Table. Fluctuating asymmetry between left and right sides for elytra length of *Stator maculatopygus* according to categories of seed biomass, fruit infestation and parasitism rate.** Results are displayed in comparison to the right elytra.
(DOCX)

## Acknowledgments

We thank the anonymous referees for their valuable comments on this study. We also thank the Federal University of Lavras and the Graduate Programme in Applied Ecology for logistic support. A. B. M. passed away before the submission of the final version of this manuscript. T. C. T. O accepts responsibility for the integrity and validity of the data collected and analyzed.

## Author Contributions

**Conceptualization:** Tamires Camila Talamonte de Oliveira, Lucas Del Bianco Faria.

**Data curation:** Tamires Camila Talamonte de Oliveira.

**Formal analysis:** Tamires Camila Talamonte de Oliveira, Angelo Barbosa Monteiro.

**Supervision:** Lucas Del Bianco Faria.

**Writing – original draft:** Tamires Camila Talamonte de Oliveira, Angelo Barbosa Monteiro.

**Writing – review & editing:** Tamires Camila Talamonte de Oliveira, Lucas Del Bianco Faria.

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
