## [Decision Letter · Decision Letter 0]

26 Aug 2020

PONE-D-20-22527

Can multitrophic interactions shape morphometry, allometry, and fluctuating asymmetry of seed-feeding insects?

PLOS ONE

Dear Dr. Tamires Camila Camila Talamonte de Oliveira

Thank you for submitting your manuscript to PLOS ONE. After careful consideration, we feel that it has merit but does not fully meet PLOS ONE’s publication criteria as it currently stands. Therefore, we invite you to submit a revised version of the manuscript that addresses the points raised during the review process.

We look forward to receiving your revised manuscript.

Kind regards,

Kleber Del-Claro, PhD

Academic Editor

PLOS ONE

Additional Editor Comments:

Dear Colleagues, one of our reviewers ma de a series of important and detailed suggestions in the manuscript. I strongly encourgae you to follow, correct or answer the criticism of this reviewer in a second round. I agree with the reviewer in most of the comments.

'TCTO thanks the Brazilian Coordination for the Improvement of Higher Education Personnel (CAPES) and the National Council for Scientific and Technological Development (CNPq) (141129/2018-2) for financial support.  LDBF thanks CAPES, CNPq (306196/2018-2), and Fundação de Amparo à Pesquisa do Estado de Minas Gerais (FAPEMIG) for financial support.'

'The author(s) received no specific funding for this work.'

Reviewers' comments:

Reviewer's Responses to Questions

**Comments to the Author**

1. Is the manuscript technically sound, and do the data support the conclusions?

Reviewer #1: Yes

Reviewer #2: Partly

2. Has the statistical analysis been performed appropriately and rigorously? 

Reviewer #1: Yes

Reviewer #2: Yes

3. Have the authors made all data underlying the findings in their manuscript fully available?

Reviewer #1: Yes

Reviewer #2: Yes

4. Is the manuscript presented in an intelligible fashion and written in standard English?

Reviewer #1: Yes

Reviewer #2: No

5. Review Comments to the Author

Reviewer #1: Please find below my comments concerning the manuscript by Oliveira et al., entitled: “Can multitrophic interactions shape morphometry, allometry, and fluctuating asymmetry of seed-feeding insects?” (PONE-D-20-22527). The authors present an interesting study regarding to multiple biotic factors that can shape the growth and development patterns of three species with similar ecological functions. This is a seminal work in this specific topic that highlight important aspects the importance of investigating several drivers factors at the same time. The text is well written, the analyses are appropriate, and I think the findings will be interesting for a broader audience. I think the study is sound and I have only some comments or recommendations.

Minor comments

L 25. What are these categories?

L 30-32. This sentence seems inconsistent and / or words are missing. Please, rewrite.

L 42. “create” could be respond through?

L 46. Are there any recent references?

L 67-69. Didn't you say that above? Line 64?

L 92: “and for the wasp: right and left wing, right and left tibia length and total body

length of A. vulgaris.“

L 92. Are there any references considering these measures for wasps?

Methods

L 108. I missed at least one sentence describing basic information about the plant such as growth form, average height, etc.

L 113-115. Were these fruits collected from the same individuals? If not, how many different individuals? Were the annual collections carried out in the same individuals? Please, provide these important details.

L 113. “seven” or eight?

L 180. Why the length and not the width?

L 238-247. The legend of Figures 4 and 5 looks similar, both in relation to S. maculatopygus.

L 262. Isn't it Figure 3? Please, see my comment above and check this figures citation on MS.

Discussion.

L 338-400. Please, detail how the great variation in allometric slopes can influence in the abundance of individuals.

L 378-379. Did the authors test this?

L 388. Maia et al. (2017).

L 411. Space before “By contrast …”

Reviewer #2: Review of manuscript #PONE-D-20-22527 “Can multitrophic interactions shape morphometry, allometry, and fluctuating asymmetry of seed-feeding insects?” by Oliveira et al.

General comments and assessment:

The manuscript (henceforth MS) assesses if multitrophic interactions might shape morphometry, allometry, and fluctuating asymmetry of seed-feeding insects in a plant species from Brazilian Cerrado. The study is innovative and brings a great advance to the understanding of how ecological interactions can influence individual performance. I really liked to read this work and I think this is a great contribution.

I can point out two major issues in the MS: there are many typos and English grammar issues. The MS is well structured, but some parts are hard to follow, or totally not understandable. The MS needs to be proofread by a native. I could provide many small issues over the MS, but it would be good to recheck the spelling and grammar.

The second point it is the lack of details in some parts of the methodology, which I highlight bellow. The main issue that I pointed out, it is that the authors did not mention how many plants they used, or how many fruits per plants they collected, which can bias the analyses, eg. spatial correlation, low variability of samples.

More specific comments:

Abstract

In my opinion the abstract needs some improvement and clarity to really reflect what was found in MS. I found some typos and issues in English grammar. Also the structure is not good, since nothing about methodology is mentioned. Further, the conclusions do not seem to follow the results. So I recommend to reword some parts and recheck English typos.

L6, and 8 – Separate city name from country with comma.

L17 – HAVE TRIED… - present perfect, not continuous.

L21-22 – You tested the effects of specific variables on morphological structures. In L22-23, you tested the effects of these specific variables on allometric patterns, fluct. asymmetry, and MORPHOLOGICAL STRUCTURE. Doesn’t morphological structure encompass these two other results (allometry and asymmetry)? What I mean it is that you used only morphology before, but later you used morphology, asymmetry, and allometry. So it would be good to standardize and be clear which parameter is different from each other and which can be encompassed by the other. In your conclusion you used only morphology; so I expect that this term encompasses the other two. If I wrong, please be clear in your statements.

L24 – Methodology is missing. I recommend you to add some words about methodology here. Maybe you can add a few words before the sentence “we tested (L. 22)…” telling time and specie used. Like: “For that, analyzing PLANT SPECIES (FAMILY) over four consecutive years, we tested how these FRUIT-RELATED TRAITS affect…

L24-27 – What is a “negative allometric pattern…”? Do you mean that the relationship between the variables fit in your models showed a negative relationship with allometric parameters? If so reword the sentence. It’s hard to follow what you want to say.

L27-28 – Again, what is the relationship between variables and

L29 – FOUND

L30-32 – When you say “besides” you give an idea of addition to “allometry”. However, later you mentioned “the most” a superlative, comparative. This is contradictory. Or both parameter, allometry and morphological, are good to evaluate, or one is better than other.

L31- “allometry, …”

L32-34 – According to your results in abstract, I can’t see that each species respond differently. You wrote that you didn’t find fluctuation asymmetry for categories of species. Only that. So I can’t say only reading the abstract that each species respond differently.

Introduction

L39-40 – change semicolons to commas. Not common in English.

L43 – change “can create” to “influence”

L44 – change “destinated” to “allocated”

L51 – add “FA” abbreviation here, when you say for the first time “fluctuating asymmetry”. Remove “fluctuating asymmetry” from L55 and L71. Use only the abbreviation instead.

L65 – For me it is a little weird mentioning your own study in a third person “Oliveira et al. …”. I would use: Recently, we observed … (Oliveira et al. 2020).

L66 – change “;” to “,”.

L69 – If you provided an abbreviation for a term (FA), please use it over the MS (there are many other parts with the term and not the abbreviation), and not the term itself.

L73-76 – I think this is not your main aim. If you see your abstract, that one is a good and appropriate aim. Provide a general view of your study and not a specific, as you did. You evaluated the effects of different interactions on the morphology of three different insects, which exploit the same plant resource. So this is bigger than what you have provided as aim. Next, you can provide all this information about the system.

L76: remove space between “Hymenoptera” and “:”

L84 – change “interesting” to “advantageous”

L86 – here you can clearly see that there is no relationship between your current aim and what you hypothesized. What is the relationship between “increase in competition and parasitism rate” and your current aim in the Introduction? The answer is none. So, because of that, you need to reword your aim.

L86-89 – For (I), you didn’t mention a specific relationship (positive, negative, or neutral) as for (II), which was “greater”, and (III), which was “reduction”. Why that? I would expect a reduction in allometric patterns.

L90 – change “explained” to “explain”

L91 – change “bruchine beetles for the Coleopteran [19,22]: pronotum” to “bruchine beetles [19,22], namely pronotum”

L92-93 – change “for the wasp right and left wing right and left tibia length and total body

length of A. vulgaris” to “for the wasp, right and left wing, right and left tibia length, and total body length of A. vulgaris”

L93-94 – You mentioned “allometric relationships and FA”, but in other parts you wrote three variables: allometry, FA, and morphology. Please, be consistent with what you are really testing and standardize you MS. See my comments in Abstract about this. Mention one term which encompasses all three, or the three terms.

L94-95 – does “seed infestation” refer to “competition” (L86)? Again, please standardize your MS and do not change the terms over the MS.

Methodology

L103-106 – the degree symbol is not correct. Remove ‘0’ before degree numbers with two digits (044).

L106 – Reference is not correctly formatted. “Tuller et al. (2015)”

L108 – Inform some characteristic of the plant species studied before saying how you assessed beetles and fruits. Is it a shrub, tree,…? What was the mean size of each plant individual? We need to know some plant traits that make your MS replicable.

L111-112 – It is not clear why you removed these data. Low abundance in the same plant, same fruit, or in general? Not clear. Because if you have low abundance in the same fruit/plant is not a reason for removing the data.

L113 – remove “Therefore”, you are not concluding something.

L113-122 – In Table 1, there is Lu-4, but not in the text. If you excluded data from 2011, why adding this information in the Table 1? Since you used 2012-2014, you can only focus on these years, and you do not need to mention Ae1 subarea, since it appeared only in 2011. Also you informed La2 in the text, but not in the Table. You need to rewrite all this part of your methodology, including 2.1 subtopic and Table 1. Remove 2011 data, and reword Ae2 and Ae3 to Ae1 and Ae2. Make further modifications accordingly to these suggestions.

L113 – How many plants did you use? How many plants per subarea? Were the same plants over months and years? How many fruits per plant?

L118 – Is it possible to give some information about fruit ripening at the moment of collection?

Table 1 – Depending on the information provided in the text, you don’t need to provide a table, since it will repeat the same information.

L1234 – remove “they were”

L128 – How did you assess seed biomass? Did you dry seeds or not? When did you do that? After how many days, weeks, months? Did you open the fruit to collect seeds or not? Please provide informations about it.

L129-130 – You have already provided full name of species with authors. Please mention only genus abbreviation and epithet.

L131 – Didn’t understand. Did you use Tuller et al. 2015 to collect these data? I can’t follow this message. It seems that you used biomass and abundance data from Tuller, and not from your study. Maybe you collected the data accoding to Tuller methodology. Please clarify that.

L131 – Wrong citation format for PlosOne.

L134 – Fig. 1 - This is not a plant species food web, but a plant-insect food web. S… tenuifolia is part of the food web. Do not need to write gray scale unless you will explain something from it.

L139 – change to … “previous studies [citation, according to PlosOne rules]”.

L146 – change “structures measures” to “structure measurements”

L147 – change to “width, and both”

L147 – change to Fig. 2AB

L149 – add space between “)[“

L151 – change to Fig. 2C

L151 – Format the citation according to Journal rules.

L155 - change to Fig. 2D

L 159 – remove space between “Hymenoptera” and “:”

L169 – change to “tested”, and to “and resource size”

L169 – Fruit infestation is competition? Be clear. Earlier you mentioned competition and infestation. Standardize your terms.

L169-170 – You mentioned the supplementary here as more details about the effects of some variables on morphometric patterns. However, these two sup. files are only formulas to calculate fruit infestation and parasitism. Also, these formulas can easily be added in the MS by: FIR=HT/SA or FIR=HT.SA-1, where FIR is the fruit infestation rate by seed-feeding insects, HT is the total abundance of herbivorous, SA is the total number of seed per fruit; and FPR=PT.(HT+PT)-1, where FPR is the fruit parasitism rate by both coleopteran and hymenopteran parasitoid present in the fruit, PT is the number total of parasitoids in the fruit, and HT is the abundance total of seed-feeding insects in the fruit (Merobruchus terani, Stator maculatopygus and Allorhogas vulgaris). If you keep the formulas in sup., please add formulas according to the order you mention the variables in the MS, i.e., first infestation, then parasitism formula.

L171-173 – change to “We defined three categories for both fruit infestation and parasitism rate (adapted from FORMATED CITATION): “Low”, from 0 to 0.30; “Medium”, from 0.31 to 0.60, and “High” from 0.61 to 1.

L175-176 – use “from” and “to” to say range of seed size, and not hyphen.

L174(169) – you used the term seed size, but you are actually using biomass. So standardize your text and say exactly what you are measuring, which seems to be biomass and not size.

L175-176 – Use only three decimals.

L178 and others – Inform which correlation, Spearman, Pearson, Kendall...

L187 – change “;” to “,”

L191 – change to “We evaluated the fluctuating asymmetry patterns of the species”

L194 – change to “We used mixed models analysis with restricted likelihood (REML),”

L196 – add comma after “i.e.”. Standardize to “fixed effects” or “fixed terms”

L198 – This paragraph is other part of mixed analysis. So, join this paragraph with the previous one.

Results

L221 – change to “lower”

L222 – Table 2 with space. Provide full name of plant species, and insects as well. Provide range for all variables or none. Change to “Total abundance” only. We already know that it is per species. Suggest LFI, MFI, HFI; LPR, MPR, HPR.

L227 – Change to “Figs.”

L230-232 – Why not comparing the slopes among categories using likelihood ratio test? It’s better than saying there is “some kind of effect”.

L248-249 – I can see a positive effect of infestation in elytra and pronotum on both beetles and pronotum and wing of the wasp. Why not for elytra and why not for the other beetle species?

L254-256 – How did you know that infestation affected negatively elytra?

L248-256 – In this case here, I’m quite sure that if CI does not overlap with slope=0 you have a significant influence of variable on insect parameter, and if it is above will have a positive and bellow a negative effect. Also these comments are to 257-265 and 266-278.

L287 – Add space after “(DA).”

L288 – remove “its”.

Discussion

L344 – format citation according to PlosOne rules.

L355-358 – This period is hard to follow. Maybe you should divide it or reword.

L356 – remove the dot after the species epithet.

L369 – format citation according to PLosOne rules. Also see L375.

L376-377 – can’t follow the sentence “of their development of the hosts”

L388 – check citation format.

L404 – what “this” refers to? Be clear.

L409 – remove comma after M. terani. Do not separate subject and verb.

L409-411 – Can’t understand these sentences. Reword it.

L412 – remove comma after species name.

L415 – What is “predated” – Do you mean “preyed upon by”?

L429 – remove “the” before “three”

L423 – incorrect use of food web. The food web is not of a plant species. The plant species is part of the food web. Reword it.

L433-434 – although their bodies scaling did not change ITS pattern - you are referring to scaling.

L434-435 – “they vary substantially in their allometric slopes at high levels of infestation and competition,”. I can’t know what is they, their,… very hard to follow.

L439 - change to “insects’ body”

Acknowledgements

If I’m not mistaken there is specific sentence for Brazilian foundation program CAPES.

FIGURES

In Fig. 1 there is the word FIGURAS at the top of the page. Change to FIGURES.

Fig 2-7 – Increase the size of number in x- and y-axis. They are very small.

Fig. 6 – You add posthoc test in your figure but you didn’t mention it in the analyzes. Did you use Tukey? Estimated marginal means? Also place the letters close to the SE bars.

Fig. 6-7 – Please do not use red color in MSs since it is not accessible for readers with reduced color vision. It would be grateful if you could change the color.

SUPPLEMENTARY

Table S1 – in M1 add space before ~

In M2 remove space after side in random effect

L26 - Change “biomass seed” to “seed biomass”

In legends of Figures and Tables provide full name of species. Abbreviate only if you mention earlier in the own legend.

6. PLOS authors have the option to publish the peer review history of their article (what does this mean?). If published, this will include your full peer review and any attached files.

Reviewer #1: No

Reviewer #2: **Yes: **Eduardo Soares Calixto

---

## [Author Response · Author response to Decision Letter 0]

26 Sep 2020

We have revised our manuscript following editor’s suggestion using a word processing program. Indeed, we agreed with most of the suggestions/comments pointed out and have answered point by point the comments raised by both reviewers. The MS English grammar has been revised, and all details about fruit collection were provided in the Materials and methods section. We hope this time expectations are met. Finally, we have organized the comments adding numbers to potential questions so we could answer each point.

The document with the specific answers were attached with the manuscript.

---

## [Decision Letter · Decision Letter 1]

21 Oct 2020

PONE-D-20-22527R1

Can multitrophic interactions shape morphometry, allometry, and fluctuating asymmetry of seed-feeding insects?

PLOS ONE

Dear Tamires Camila Talamonte de Oliveira

Thank you for submitting your manuscript to PLOS ONE. After careful consideration, we feel that it has merit but does not fully meet PLOS ONE’s publication criteria as it currently stands. Therefore, we invite you to submit a revised version of the manuscript that addresses the points raised during the review process.

We look forward to receiving your revised manuscript.

Kind regards,

Kleber Del-Claro, PhD

Academic Editor

PLOS ONE

Additional Editor Comments (if provided):

Dear Author, please do the needed corrections pointed by one reviewer and I will give the final decision without the need of a third round.

Reviewers' comments:

Reviewer's Responses to Questions

**Comments to the Author**

1. If the authors have adequately addressed your comments raised in a previous round of review and you feel that this manuscript is now acceptable for publication, you may indicate that here to bypass the “Comments to the Author” section, enter your conflict of interest statement in the “Confidential to Editor” section, and submit your "Accept" recommendation.

Reviewer #1: All comments have been addressed

Reviewer #2: (No Response)

2. Is the manuscript technically sound, and do the data support the conclusions?

Reviewer #1: Yes

Reviewer #2: Yes

3. Has the statistical analysis been performed appropriately and rigorously? 

Reviewer #1: Yes

Reviewer #2: Yes

4. Have the authors made all data underlying the findings in their manuscript fully available?

Reviewer #1: No

Reviewer #2: Yes

5. Is the manuscript presented in an intelligible fashion and written in standard English?

Reviewer #1: Yes

Reviewer #2: Yes

6. Review Comments to the Author

Reviewer #1: Please find below my comments concerning the manuscript by Oliveira et al., entitled: “Can multitrophic interactions shape morphometry, allometry, and fluctuating asymmetry of seed-feeding insects?” (PONE-D-20-22527). The authors presented an excellent review of the MS (point by point) which improved the quality of MS. In addition, there was a systematic review of English grammar. I am satisfied with the final version and would like to consider publishing this MS in PlosOne journal.

Reviewer #2: Review #2 of manuscript #PONE-D-20-22527 “Can multitrophic interactions shape morphometry, allometry, and fluctuating asymmetry of seed-feeding insects?” by Oliveira et al.

Authors did a great job. The MS is very clear now. There still are some grammatical errors. I identified some, and I suggest again a careful check of English.

L112-113 – Do not use abbreviation for species name when starting a period. So provide full genus name.

L113, 209 and elsewhere – after “i.e.” add a comma

L220 – Add space between period and “The”.

L255 – I think there is a typo when informing the supplementary. After S2 there is a hyphen and comma, and then S4. Check it out.

L274-280 – Very long sentence. Please divide it in two at least.

L281 – Remove space after scale.

L289 – I believe after MP is a semicolon and not “:”

L296 – Change to “…seed). No variations…”

L301 – Remove “(medium seed)”, you have already mentioned it in 295. Then use the abbreviation.

L395 – I think “growths” is not correct. Do you mean: structure growth?

L406 – Change to [68,69]. Thus,… The dot is in a wrong place.

L415 – I did not understand this sentence. “to changes in the size of their resource,”. First I think it is “change”. And I don’t what is a change in size of a resource. Would it be amount of resource. Please check it out.

L419 – Use abbreviations constantly over the MS. Do not need to write the words again.

L426-429 – This sentence is not well written and clear. I suggest “Some studies have suggested that body size is more sensitive to environmental disturbances than FA, since body size of individuals can present a greater plasticity than FA [75–77].”. Please also check what you mentioned before has the same meaning now.

L453 – change to “are limiting factors”

L466-470. I found very repetitive theses sentences, and also you suggest/recommend studies in all of them. I recommend you to remove the last two sentences or reword them: “We recommend the use of insects’ body size and morphological structures in studies of food web. However, to clarify and better understand these approaches, more studies are necessary.”

7. PLOS authors have the option to publish the peer review history of their article (what does this mean?). If published, this will include your full peer review and any attached files.

Reviewer #1: **Yes: **Diego Anjos

Reviewer #2: No

---

## [Author Response · Author response to Decision Letter 1]

21 Oct 2020

We thank the reviewers for the valuable comments on the manuscript. We have revised our manuscript following editor’s suggestion using a word processing program. Indeed, we agreed with most of the suggestions/comments pointed out and have changed point by point according to reviewer #2 corrections.

---

## [Editor Report · Decision Letter 2]

23 Oct 2020

Can multitrophic interactions shape morphometry, allometry, and fluctuating asymmetry of seed-feeding insects?

PONE-D-20-22527R2

Dear Dr. Tamires Camila Talamonte de Oliveira,

We’re pleased to inform you that your manuscript has been judged scientifically suitable for publication and will be formally accepted for publication once it meets all outstanding technical requirements.

Kind regards,

Kleber Del-Claro, PhD

Academic Editor

PLOS ONE
---

## [Editor Report · Acceptance letter]

27 Oct 2020

PONE-D-20-22527R2 

Can multitrophic interactions shape morphometry, allometry, and fluctuating asymmetry of seed-feeding insects? 

Dear Dr. Oliveira:

I'm pleased to inform you that your manuscript has been deemed suitable for publication in PLOS ONE. Congratulations! Your manuscript is now with our production department. 

Kind regards, 

on behalf of

Dr. Kleber Del-Claro 

Academic Editor

PLOS ONE